

# A comparative study of the capability of MSCs isolated from different human tissue sources to differentiate into neuronal stem cells and dopaminergic-like cells

Nidaa A. Ababneh[1,*], Ban Al-Kurdi[1,*], Fatima Jamali[1] and Abdalla Awidi[1,2,3]

[1] Cell Therapy Center (CTC), the University of Jordan, Amman, Jordan
[2] Hemostasis and Thrombosis Laboratory, School of Medicine, the University of Jordan, Amman, Jordan
[3] Department of Hematology and Oncology, Jordan University Hospital, Amman, Jordan
[*] These authors contributed equally to this work.

Corresponding authors
Nidaa A. Ababneh,
nidaaanwar@gmail.com
Abdalla Awidi,
abdalla.awidi@gmail.com

## ABSTRACT

**Background.** Neurodegenerative diseases are characterized by progressive neuronal loss and degeneration. The regeneration of neurons is minimal and neurogenesis is limited only to specific parts of the brain. Several clinical trials have been conducted using Mesenchymal Stem Cells (MSCs) from different sources to establish their safety and efficacy for the treatment of several neurological disorders such as Parkinson's disease, multiple sclerosis and amyotrophic lateral sclerosis.

**Aim.** The aim of this study was to provide a comparative view of the capabilities of MSCs, isolated from different human tissue sources to differentiate into neuronal stem cell-like cells (NSCs) and possibly into dopaminergic neural- like cells.

**Methods.** Mesenchymal stem cells were isolated from human bone marrow, adipose, and Wharton's jelly (WJ) tissue samples. Cells were characterized by flow cytometry for their ability to express the most common MSC markers. The differentiation potential was also assessed by differentiating them into osteogenic and adipogenic cell lineages. To evaluate the capacity of these cells to differentiate towards the neural stem cell-like lineage, cells were cultured in media containing small molecules. Cells were utilized for gene expression and immunofluorescence analysis at different time points.

**Results.** Our results indicate that we have successfully isolated MSCs from bone marrow, adipose tissue, and Wharton's jelly. WJ-MSCs showed a slightly higher proliferation rate after 72 hours compared to BM and AT derived MSCs. Gene expression of early neural stem cell markers revealed that WJ-MSCs had higher expression of Nestin and PAX6 compared to BM and AT-MSCs, in addition to LMX expression as an early dopaminergic neural marker. Immunofluorescence analysis also revealed that these cells successfully expressed SOX1, SOX2, Nestin, TUJ1, FOXA2 and TH.

**Conclusion.** These results indicate that the protocol utilized has successfully differentiated BM, AT and WJ-MSCs into NSC-like cells. WJ-MSCs possess a higher potential to transdifferentiate into NSC and dopaminergic-like cells. Thus, it might indicate that this protocol can be used to induce MSC into neuronal lineage, which provides an additional or alternative source of cells to be used in the neurological cell-based therapies.

## INTRODUCTION

Mesenchymal stem cells (MSCs) are a population of cells characterized by their great regenerative capacity and mutlipotent differentiation potential into multiple cell lineages (*Ullah, Subbarao & Rho, 2015*). These cells can be easily isolated from different tissue sources with minimal invasive procedures (*Ullah, Subbarao & Rho, 2015*). MSCs have been isolated from adipose tissue, bone marrow, Wharton jelly, dental pulp and umbilical cord blood. MSCs have the potential to differentiate into adipogenic, osteogenic and chondrogenic lineages. Some studies have reported the ability of these cells to cross lineage commitment and to differentiate into endodermal and ectodermal cell lineages (*Ullah, Subbarao & Rho, 2019*; *Orbay, Tobita & Mizuno, 2012*; *Ullah, Subbarao & Rho, 2015*). Additionally, MSCs are hypoimmunogenic and have immunosuppressive properties. All of these characteristics and the fact that MSCs are not burdened by ethical issues, vector integration, genomic instability, inefficient generation and tumorigenic capacity associated with embryonic stem cells (ESCs) and induced pluripotent stem cells (iPSCs), makes them an attractive choice for tissues engineering and cell replacement therapies (*Ullah, Subbarao & Rho, 2019*; *Musiał-Wysocka, Kot & Majka, 2019*; *Wang, Yuan & Xie, 2018*; *Medvedev, Shevchenko & Zakian, 2010*).

Parkinson's disease (PD) is a neurodegenerative disorder characterized by the loss of dopaminergic neurons, resulting in an impairment of the motor function (*Alexander, 2004*). The loss of these cells makes the PD an attractive model for cell replacement therapies. No specific treatment is currently available to treat PD patients. Surgical therapies as well as different pharmacological treatments have been utilized to relieve some of the PD symptoms (*Alexander, 2004*; *Dauer & Przedborski, 2003*; *Fu et al., 2015*). However, treatments usually fail after a while, due to the progressive nature of the disease. Searching for a more effective therapeutic strategy is essential to hinder the progression of dopaminergic neurons degeneration.

Several clinical trials have been conducted to assess the safety and efficacy of using MSCs for the treatment of graft *versus* host disease, heart failure, bone and cartilage diseases, neurodegenerative and spinal cord injuries (*Musiał-Wysocka, Kot & Majka, 2019*; *Ul Hassan, Hassan & Rasool, 2009*; *Ullah, Subbarao & Rho, 2015*).

Different differentiation protocols have been utilized to direct MSCs towards the neuronal lineage. Cell culture media supplemented with FGF2, EGF, BMP-9, retinoic acid, and heparin have been used to induce MSCs derived from adipose tissue to cholinergic and dopaminergic neuronal-like cells (*Marei et al., 2018*). Additionally, the soluble factors sonic hedgehog (SHH), fibroblast growth factor 8 (FGF8), and basic fibroblast growth factor (bFGF) along with final treatment with BDNF neurotrophic factor have been successfully used to generate functional dopaminergic neurons from WJ, ASC, UC and olfactory Mesenchymal Stem Cells (*Boroujeni & Gardaneh, 2017*; *Khademizadeh et al., 2019*; *Ul Hassan, Hassan & Rasool, 2009*; *Yang et al., 2013*). Choroid Plexus Epithelial Cell,

**Table 1** Summary of the different protocols utilized for Mesenchymal stem cell differentiation to Dopaminergic neurons.

| Protocol | Reference |
| --- | --- |
| Cell culture media supplemented with FGF2, EGF, BMP-9, retinoic acid, and heparin | *Marei et al. (2018)* |
| choroid plexus epithelial cell-conditioned medium (CPEC-CM) | *Boroujeni et al. (2017)* |
| Cell culture media supplemented with sonic hedgehog (SHH), 100 ng/mL fibroblast growth factors (FGF)-8 and 50 ng/mL bFGF | *Khademizadeh et al. (2019)* |
| Inducible lentivirus-mediated hGDNF gene in MSCs | *Yang et al. (2013)* |

mesencephalic glial-cell, PA6 stromal cells-derived conditioned media have also been used to induce dopaminergic differentiation on different stem cell types (*Boroujeni et al., 2017*) Table 1. Transduction of MSCs with transcription factors required for dopaminergic differentiation such as LMX1, NTN and GDNF using lentiviral or retroviral vectors have been proved to be an efficient way to enhance the differentiation potential of MSCs towards the dopaminergic lineage (*Barzilay et al., 2009*; *Ul Hassan, Hassan & Rasool, 2009*; *Yang et al., 2013*) Table 1.

Most of the previous studies have shown variations in differentiation potential of different types of MSCs into neuronal lineage. Hence, the aim of this study was to assess and compare the neural dopaminergic differentiation capacity of MSCs isolated from adipose tissue, bone marrow, and Whartons' jelly. Such findings might assist in choosing the appropriate cell source to be utilized in cell replacement and neural regenerative therapies.

## MATERIALS AND METHODS

### Isolation and characterization of MSCs from different tissue sources

This study was conducted after obtaining an International Review Board (IRB/7/2019) approval at the University of Jordan/Cell Therapy Center (CTC). Samples were collected after all donors gave written informed consents.

Six samples from six different donors for each tissue type were used to isolate MSCs: Adipose tissue (mean = $32.6 \pm 5.4$; three males and three females), Bone marrow (mean = $38.2 \pm 7.0$; three males and three females) and Wharton's Jelly tissue (mean = $31.2 \pm 3.5$; 6 females). The isolation of stem cells was performed according to the protocols utilized by following protocols, respectively (*Bunnell et al., 2008*; *Gnecchi & Melo, 2009*; *Ranjbaran et al., 2018*). The isolated cells obtained from these tissues were cultured in Minimum Essential Medium alpha (α-MEM; Gibco) supplemented with 5% pooled human platelet lysate (hPL), 1% Antibiotic-Antimycotic (Gibco) and 1% Glutamax (Gibco). Cells at 70%–80% confluence were expanded until passages 1–3. Then cells were either used for further experiments or stored in liquid nitrogen.

## Characterization of MSCs
### Flow cytometry
Cells at passage 3 and 70% confluency were utilized for MSC surface markers assessment using Human MSC Analysis Kit (BD, USA). Briefly, cells were detached with 1X TryplE (Gibco) and washed twice with FACS buffer (PBS, 1% FBS). After that, cells were resuspended in FACS buffer and the concentration was adjusted to 1 X $10^6$ cells/ml. Aliquots of 100 µl from the cell suspension were placed in test tubes and incubated for 30 min in the dark with fluorochrome conjugated antibodies against CD-44, CD-105, CD-73, CD-90 and a negative cocktail mix according to the manufacturer instructions. Cells were then centrifuged at 300xg for 5 min and resuspended in 500 µl FACS buffer. Analysis was performed using FACSCanto$^{TM}$ (BD) and the data were analyzed using Diva software.

### Multilineage differentiation
Adipogenic differentiation was performed using StemPro Adipogenesis differentiation media (Invitrogen, Waltham, MA, USA) for 14 days. Cells were washed twice with PBS, fixed and stained with Oil red O stain to confirm the adipogenic differentiation potential. StemPro Osteogenic differentiation kit (Invitrogen, USA) was used to induce ASC differentiation towards the urothelial lineage. After 21 days in culture, cells were washed, fixed and stained with Alizarin red stain to verify osteogenic differentiation. Cells under normal culture conditions were used as a negative control.

## Cell proliferation analysis
MSCs were seeded onto 96-well plates at a unified seeding density of 5,000 cells/well and cultured under normal conditions for three days. MTT (3-(4, 5-dimethylthiazolyl2)-2, 5-diphenyltetrazolium bromide (ATCC® 301010K) was used to measure the cell proliferation rate after 24, 48 and 72 h according to the manufacturer's instructions.

## Neural induction
Cells were seeded on Matrigel coated six well plates and coverslips, and in the following day medium was changed into neural induction media consisting of Dulbecco's modified Eagle's medium F12 (DMEM/F12; Gibco) supplemented with 3% Knockout Serum Replacement (KSR, Gibco), 1% Glutamax (Gibco, Waltham, MA, USA), 1% non-essential amino acid (NEAA, Gibco), 4 ng/mL basic fibroblast growth factor (bFGF; Peprotech), 10 µM SB431542 (Sigma), and 0.5 µM LDN193289 (Sigma) for 8 days. Following, cells were passaged at 1:3 split ratio and media was switched every other day as the following: Day 7 and 8 75% neural induction media and 25% of Neuroabasal media consisting of 0.5% B27, 0.5% N2, 100 nM LDN, 100 ng/mL SHH C24II, 2 µM Purmorphamine, 100 ng/mL FGF8a, 3 µM CHIR-99021. Day 9 and 10 50% neural induction media and 50% Neuroabasal media 0.5% B27, 0.5% N2 100 nM LDN 3 µM CHIR-99021. Day 11 and 12 25% neural induction media and 75% of Neuroabasal media 100 nM LDN 3 µM CHIR-99021. Day 13 onward 3 µM CHIR-99021, 10 ng/ µl BDNF, 10 ng/ µl GDNF, 1 ng/mL TGFb3, 10 µM DAPT, 200 µM Ascorbic Acid and 500 µM db-cAMP.

**Table 2  qPCR Primer sequence.**

| Gene Name | Forward (5 → 3) | Reverse (5 → 3) |
|---|---|---|
| GAPDH | CCTGTTCGACAGTCAGCCG | CGACCAAATCCGTTGACTCC |
| NKX6.1 | ATTCGTTGGGGATGACAGAG | CCGAGTCCTGCTTCTTCTTG |
| Nestin | AGAAACAGGGCCTACAGAGC | GAGGGAAGTCTTGGAGCCAC |
| Sox-2 | TAGAGCTAGACTCCGGGCGAT | TTGCCTTAAACAAGACCACGAAA |
| Pax6- | CGGAGTGAATCA GCTCGGTG | CCGCTTATACTGGGCTATTTTGC |
| TUJ | GCGAGATGTACGAAGACGAC | TTTAGACACTGCTGGCTTCG |
| LMX1A | AGGAAGGCAA GGACCATAAGC | ATGCTCGCCTCTGTTGAGTTG |

## Gene expression analysis

Total RNA was isolated on day 30 using RNeasy mini kit (Qiagen, Valencia, CA, USA) according to the manufacturer's instructions. cDNA was synthesized using PrimeScript RT Master Mix (TaKaRa, Lexington, MA, USA). Quantitative real-time PCR (qPCR) was preformed using iQSYBR mix (BioRad, USA) according to the manufacturer's protocol and using specific forward and reverse primers listed in Table 2. qPCR results were analyzed using the $\Delta \Delta$CT relative quantification method.

## Immunofluorescence staining of neuronal markers

After 7 and 30 days of induction, cells on coverslips were fixed in 4% formaldehyde for 15 min and permeablized with 1X PBS containing 0.1% TritonX-100 for 5 min. To prevent nonspecific binding, cells were incubated with blocking solution consisting of 10% normal goat serum and 0.3% TritonX-100 (v/v) in 1X PBS for 60 min. Cells were then incubated with the primary antibodies against SOX2, SOX1, TUJ1, Nestin, PAX6, and TH diluted in blocking buffer overnight at 4 °C. Subsequently cells were incubated with appropriate secondary antibodies at 4 °C in the dark and then DAPI staining and mounting onto microscope slides. Cells were imaged using AxioObserever Z1 microscope (Zeiss, Germany).

## Statistical analysis

All the experiments were done at least three times and statistical analysis was performed using GraphPad Prism (Version 6). The data were presented as the mean ± standard error of the mean (SEM). Statistical differences were calculated using One-Way Analysis of Variance (ANOVA 1) and Post-hoc test for comparison between groups. Differences were considered significant at (* $P < 0.05$, ** $P < 0.01$, *** $P < 0.001$).

# RESULTS

## Isolation and characterization of ADSCs

On the third day of primary culture, cells with fibroblastic morphology were adhered to the tissue culture plate and became confluent within 14 days of initial plating. To validate the stemness of the isolated cells, MSCs from different sources were transdifferentiated into the adipogenic and osteogenic cell lineages. Cells induced with adipogenic media for 14 days exhibited intracellularly localized lipid droplets stained with Oil red O (ORO), which

were absent in the negative control (Fig. 1A). Following 14 days of osteogenic induction, cells exhibited flattened and more elongated morphology with extracellular deposits. The presence of extracellular calcium phosphate deposits was confirmed with Alizarin Red stain (ARS), which were absent in the uninduced negative controls (Fig. 1B). Thus, the cells were successfully differentiated into osteoblasts and pre-adipocytes, confirming the multi-potency of these cells. Flow cytometry analysis showed positive expression of the following MSCs markers: CD-90, CD-105, CD-73 and minimal to no expression of markers in the negative cocktail (Fig. 2A).

## Analysis of cellular proliferation

Cellular proliferation was assessed using MTT assay to compare the proliferation rate between MSCs-derived from AT, BM and WJ. After 48 h, no significant difference in the proliferation rate was observed between the analyzed cell types. However, WJ derived MSCs showed a significantly higher proliferation after 72 h compared to BM and AT-derived MSCs ($P = 0.009$) (Fig. 2B).

## Morphological analysis of MSCs induced towards the neuronal lineage

The morphology of MSCs induced towards the dopaminergic lineage demonstrated several alterations throughout the differentiation process. Initially, the three different MSC types-derived from AT, BM and WJ adhered to the culture flask and exhibited a spindle-shaped morphology. Following splitting and prolonged culture, cells began to change in shape and acquire a more spherical appearance, obtaining a neural-like morphology with the appearance of cellular processes (Fig. 3A). This change was more distinguishable in WJ-MSC compared to AT-MSC and BM-MSCs.

## Molecular analysis of neuronal induction

Quantitative real-time PCR was used to evaluate the relative expression of the following neuronal genes: β Tubulin III (TUJ1), Nestin, NKX6.1, SOX2, PAX6 and LMX at the transcriptional levels. Expression levels of the same target genes in undifferentiated original MSCs were considered as the baseline level. The expression of TUJ1 gene, a neuronal marker of immature neurons, was upregulated in all differentiated MSCs without any significant differences between different MSC sources (Fig. 3B). Nestin gene expression, a marker of neural progenitors, revealed a significant upregulation in WJ compared to AT (*$P < 0.05$) and BM (**$P < 0.01$) derived MSCs ((Fig. 3B). On the other hand, NKX6.1 is also an early neuronal progenitor marker, was upregulated without any statistical differences between the induced MSC types. SOX2 is a marker for early and intermediate progenitor neuronal cells (Fig. 3B). Levels of expression of this gene in WJ derived MSCs was significantly higher compared to BM derived cells (*$P < 0.05$). Additionally, PAX6 an intermediate progenitor marker was significantly upregulated in WJ derived MSCs compared to AT and BM derived MSCs (****$P < 0.0001$). Additionally, we analyzed the expression of LMX1a (LIM homeobox transcription factor 1, alpha) gene associated with the differentiation towards dopaminergic neurons, and we found that WJ derived MSCs expressed significantly higher

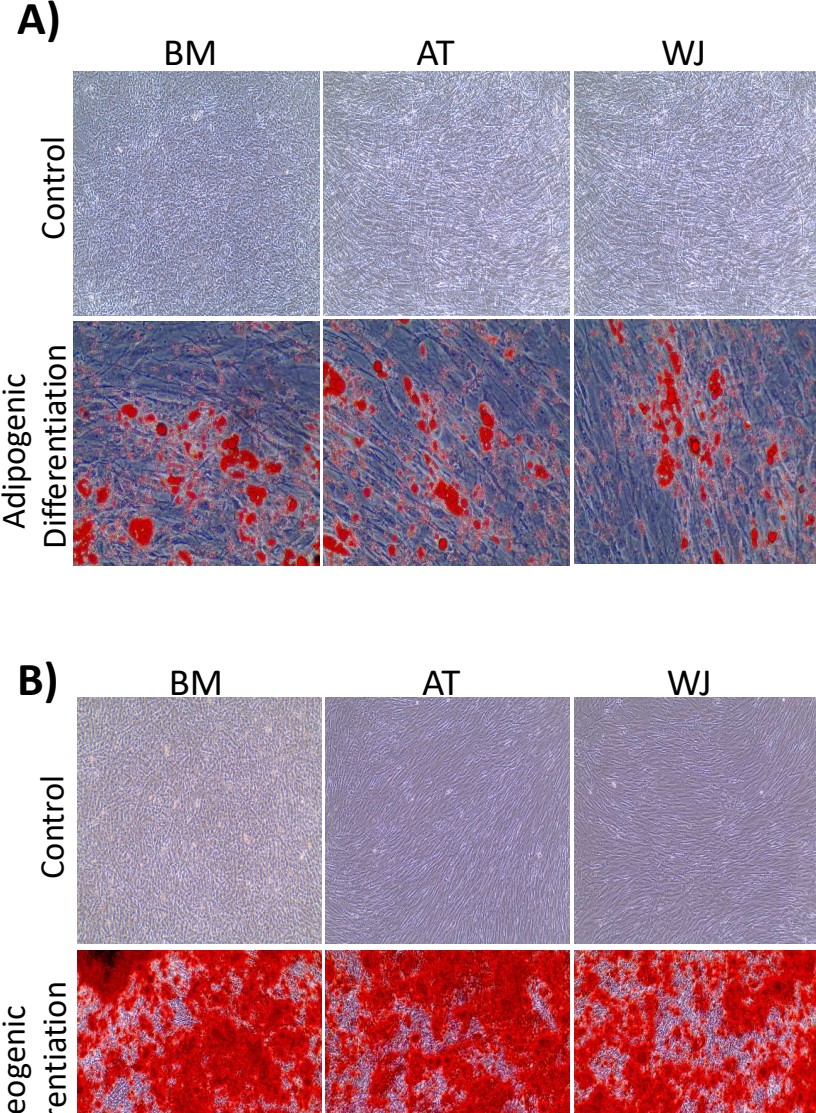

**Figure 1 Characterization of MSCs by multilineage differentiation analysis.** (A) BM, AT and WJ tissue derived MSCs after 14 days of adipogenic differentiation, showing internal lipid droplets following staining with Oil Red O, which were absent in the undifferentiated controls. (B) MSCs after 14 days of osteogenic differentiation, showing mineral deposition after staining with Alizarin Red stain (ARS), which were absent in the undifferentiated controls.

levels compared to BM and AT derived MSCs (**$P < 0.01$), which suggests that our protocol could direct this types of MSCs towards a dopaminergic-like phenotype (Fig. 3B).

Gene expression results support the morphological changes seen under the microscope and support the idea that our differentiation protocol successfully generated neural stem/progenitor-like cells from MSCs derived from BM, AT and WJ.

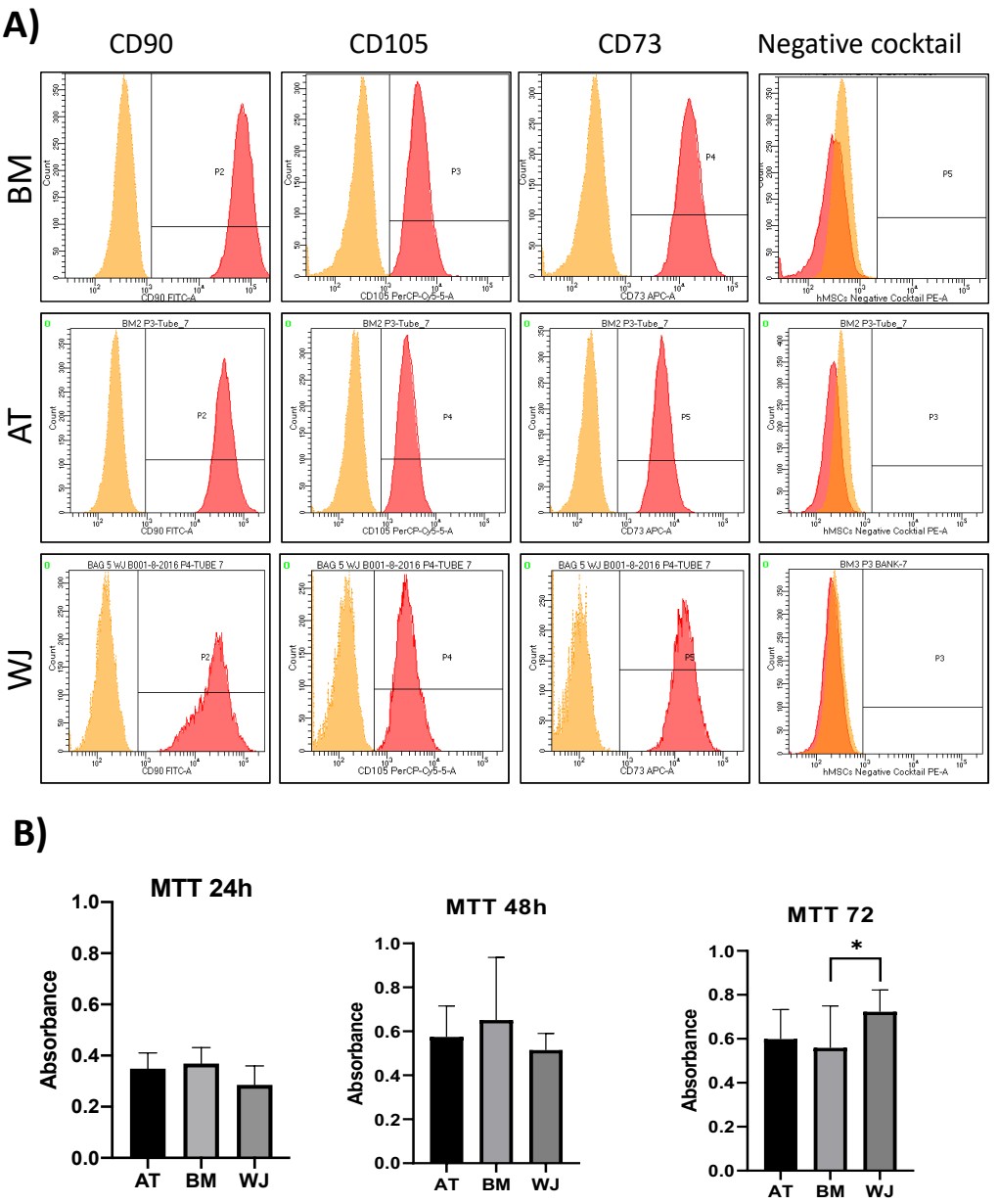

**Figure 2** **Characterization of MSCs by flow cytometry and comparison of proliferation potential.** (A) Flow cytometry analysis of MSCs showed that cells were positive for MSC markers CD-90, CD-105, CD-73 and negative for the negative cocktail mix. $N = 18$ (B) Proliferation analysis of MSCs from BM, AT and WJ samples. MTT assay was performed on cells cultured for 24, 48 and 72 h. $N = 18$, $\star$ $P < 0.05$.

## Assessment of neuronal induction by immunofluorescence

Since the morphological and gene expression results revealed a successful differentiation into the neuronal-like lineage, we confirmed the expression of NESTIN, TUJ1, SOX1, SOX2 and PAX6 by immunofluorescent staining. All of these early neuronal progenitor markers were expressed in all type of induced MSCs (Figs. 4 & 5). However, a more prominent expression of these markers in WJ-derived MSCs was detected compared to AT and BM

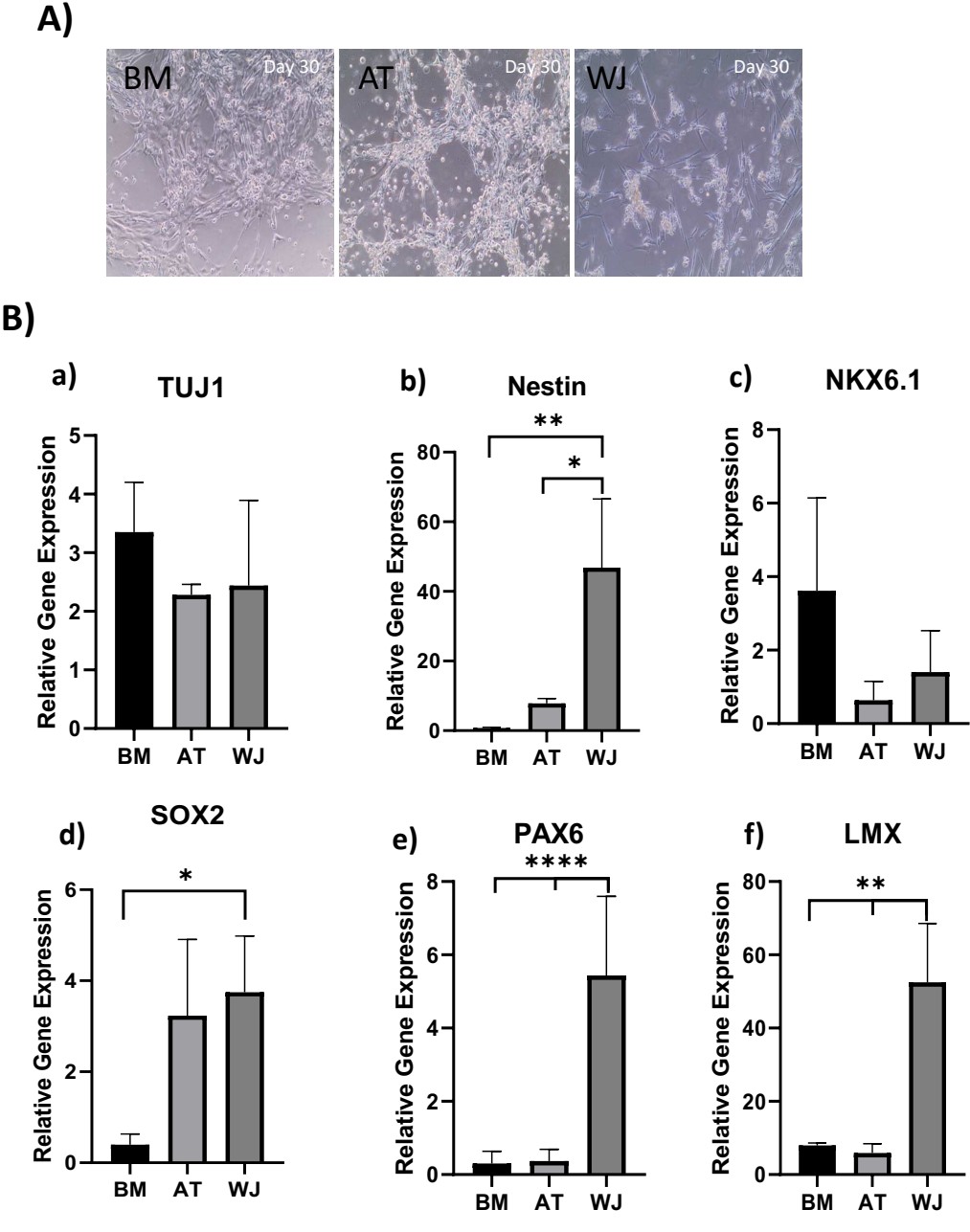

**Figure 3** **Induction of MSCs into neural stem cell-like cells utilizing dual SMAD inhibition.** (A) Phase-contrast images of BM, AT and WJ MSCs, respectively, showing the morphological changes after 30 days in culture. Scale bar: 100 μm. (B) Quantitative real-time PCR was used to assess the expression of TUJ1, Nestin, NKX6.1, SOX2, PAX6 and LMX genes on cultured neurons. Relative gene expression of each gene was normalized to the expression of GAPDH housekeeping gene. Data represents means ± SEM of three independent experiments. * $P < 0.05$, ** $P < 0.01$, *** $P < 0.001$.

derived MSCs. Additionally, cells were stained for TH (Tyrosine Hydroxylase) and FOXA-2 (Forkhead box protein A2) as markers for differentiation towards the dopaminergic lineage. WJ-MSCs showed higher expression levels compared to the other sourdes, which might

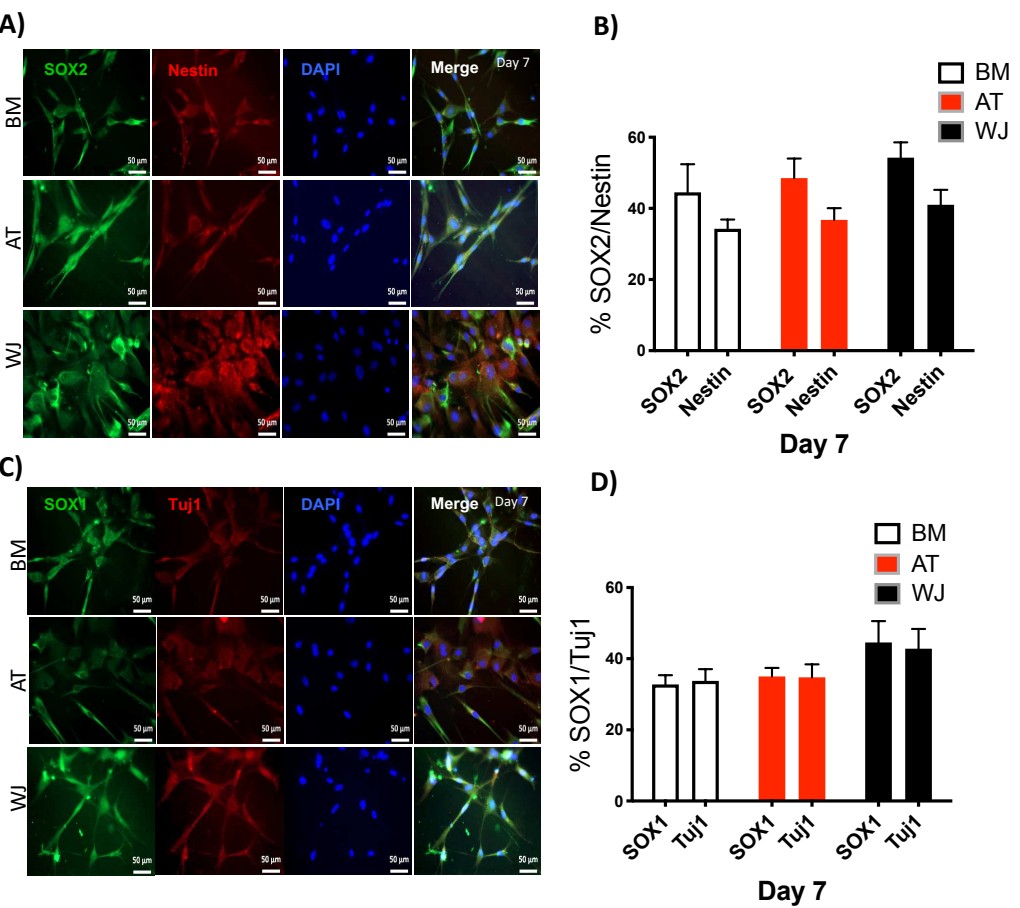

**Figure 4 Immunofluorescence analysis of neuronal markers at day 7.** Neural stem cells differentiated from MSCs and cultured in neuronal induction media were analyzed by immunofluorescence staining on day 7 for the expression of neural stem/progenitor protein markers (A) SOX2 and Nestin. Scale bar = 50 μm. (B) Semi-quantitative analysis of SOX2 and Nestin representing the percent of positive cells. (C) SOX1 and TUJ1, Scale bar = 50 μm (D) Semi-quantitative analysis of SOX1 and TUJ1 immunofluorescence representing the percent of positive cells. All experiments were repeated at least three independent times.

suggest that WJ represents a more efficient cell source for neuronal cell differentiation (Fig. 6).

## DISCUSSION

The regeneration of neurons following injury is minimal and neurogenesis is limited to specific parts of the brain (*Hess & Borlongan, 2008*). Several clinical trials have been conducted using MSCs from different sources to establish their safety and efficacy for the treatment of many neurological disorders such as Parkinson's disease, multiple sclerosis and amyotrophic lateral sclerosis (*Boroujeni & Gardaneh, 2017*; *Karussis et al., 2010*; *Syková et al., 2017*). *In vitro* differentiation studies utilizing MSCs isolated from different tissues have shown variable proliferation and differentiation potential (*Alizadeh et al., 2019*;

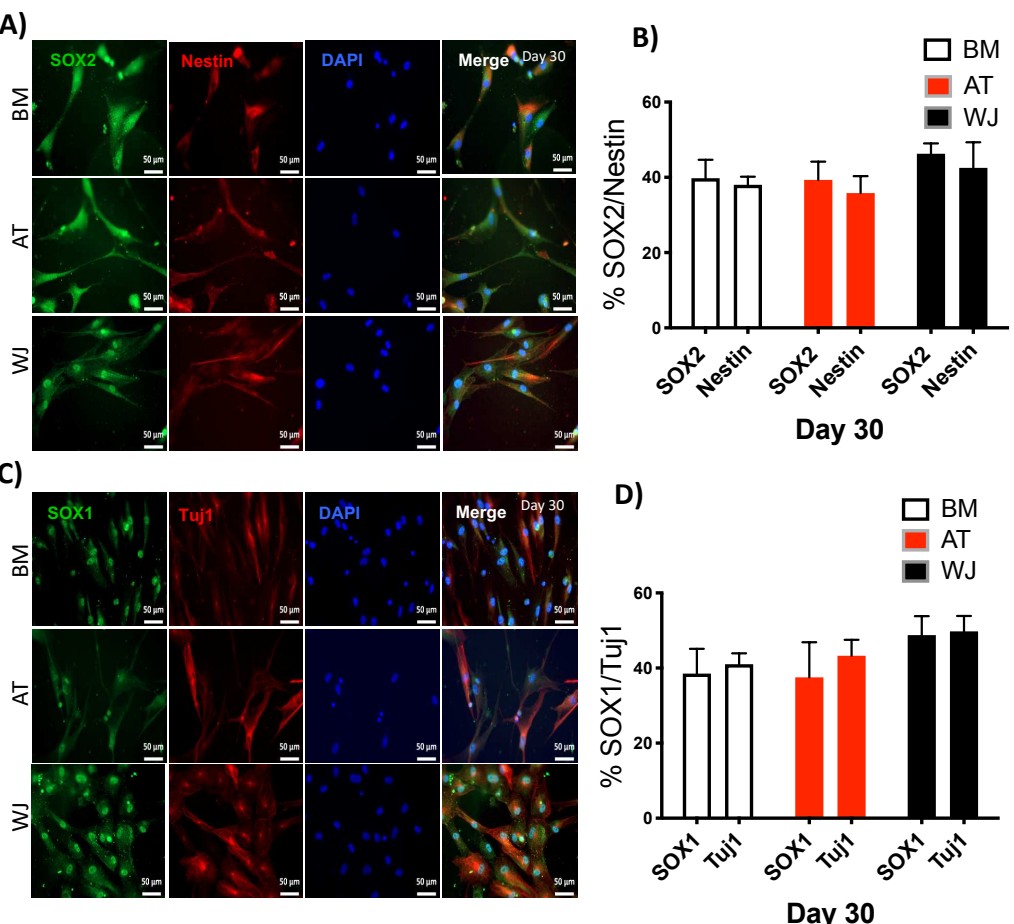

**Figure 5** **Immunofluorescence analysis of neuronal markers at day 30.** MSCs derived from bone marrow, adipose and Wharton's jelly tissue cultured in neuron induction media were analyzed on day 30 for the expression of neural stem/progenitor protein markers (A) SOX2 and Nestin. Scale bar = 50 mm. (B) Semi-quantitative analysis of SOX2 and Nestin immunofluorescence representing percentage of positive cells relative to negative control. (C) SOX1 and TUJ1, Scale bar = 50 mm (D) Semi-quantitative analysis of SOX2 and Nestin immunofluorescence representing percentage of positive cells. All experiments were repeated at least three times.

*Balasubramanian et al., 2013*; *Datta et al., 2011*; *Urrutia et al., 2019*). These variabilities can significantly impact the clinical outcome. Accordingly, there is a need to provide a clear overview and comparison on the neuronal differentiation potential of MSCs isolated from different sources. Thus, the aim of this study was to provide a comparative view of the capabilities of MSCs isolated from different human tissue sources, to differentiate into neuronal stem cell-like cells and dopaminergic-like cells. The data described here shed the light on the most appropriate MSC source of to be used therapeutically in neural regenerative therapies.

Here we confirm that MSCs isolated from adipose tissue, bone marrow, and Wharton's jelly express similar surface markers and they are capable of undergoing multilineage differentiation. The proliferative capacity of MSCs appear to be similar across the three

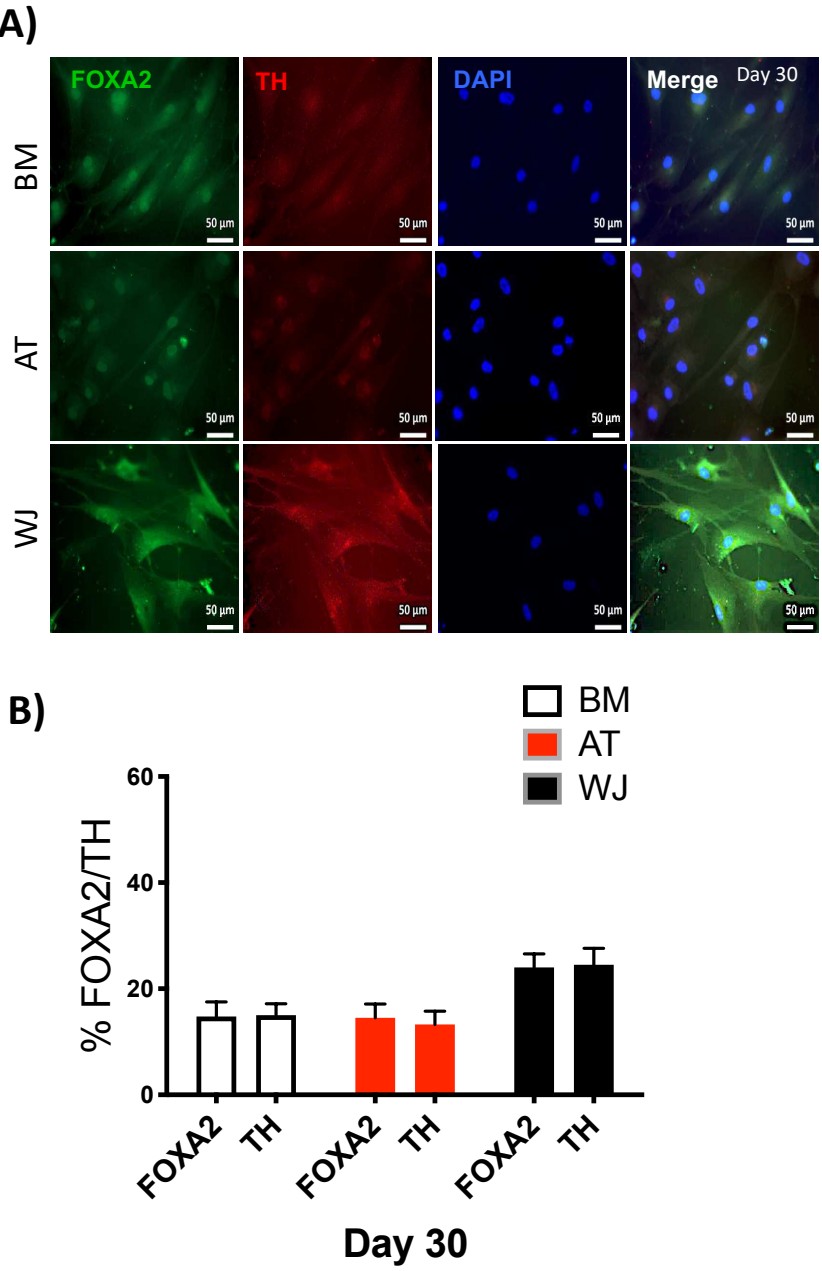

**Figure 6** **Immunofluorescence analysis of mature neuronal markers at day 30.** The expression of dopaminergic markers (A) FOXA2 and Tyrosine hydroxylase was analyzed by immunofluorescence staining on cells cultured for 30 days in neuronal induction media. Scale bar = 50 mm. (B) Semi-quantitative analysis of FOXA2 and TH immunofluorescence representing the percentage of positive cells. All experiments were repeated at least three times.

different types of MSCs with minor variations. These variations can result from culture heterogenicity, and different proportions of self-renewing cells in comparison to lineage-specific cells in different tissues. Previous studies have reported different proliferative capabilities of MSCs. Urrutia et al. reported higher proliferation rate of AT compared to

WJ and BM-derived stem cells (*Urrutia et al., 2019*). On the other hand Hu et al. reported results that contradict the above, in which WJ had higher proliferation rate compared to AT (*Hu et al., 2013*). Other studies have also reported varying results (*Aliborzi et al., 2016*; *Heidari et al., 2013*). Such variabilities might be attributed to differences in isolation and culture methods or differences due to age, sex and health status of the samples donors. The assay of choice used to measure the proliferation capacity plays a critical contribution as well. Such results are of importance for the selection of an appropriate tissue source to derive MSCs for the use in cell-based therapies, which are required in sufficient numbers in a limited time to achieve effective clinical outcomes. Hence, further rigorous analysis must be conducted to clearly identify differences in proliferation capacities, with matched large samples, and utilizing unified consistent methods.

Studies in animal models as well as human cell lines have shown that bone morphogenetic protein (BMP) and the Activin/Nodal pathway play a significant role in neural development of embryos as well as neuronal differentiation of different types of stem cells (*Park et al., 2017*; *Wattanapanitch et al., 2014*). Several differentiation studies have demonstrated the synergistic inhibition of those two pathways utilizing a small molecules cocktail such as SB 431542, Noggin, LDN 193189 to induce the cells towards the neural progenitor fate that can then be differentiated to a more mature neural cell type (*Park et al., 2017*; *Pauly et al., 2018*; *Wattanapanitch et al., 2014*).

Here, we utilized a combination of small molecules to direct the differentiation of MSCs towards the neural lineage in a serum-free environment. Small molecules are relatively cheap, stable and have high penetrating capability. Briefly, we employed dual-SMAD inhibition during the initial stage of differentiation. Dual SMAD inhibition was achieved by adding SB-431542 as a TGF-$\beta$ inhibitor and LDN-193189 as a BMP-inhibitor to induce the neural lineage. Thus, revealed that MSCs from different sources are capable of generating neural stem cell (NSC)-like cells. Following 7 days of the initial induction, MSCs derived from different human tissue-sources changed their morphology into spindled neuronal-like shape. Furthermore, we assessed the expression of a group of NSC markers, including Nestin, Tuj1, Pax6, Sox1, Sox2. Our results point towards the ability of MSCs to differentiate into neural stem cell-like phenotype. These cells are of interest to provide an intermediate neural cell source that can be further differentiated into a more mature state. However, it is important to note that these cells must be further characterized for their purity and the ability to express a comprehensive panel of NSC markers. Furthermore, the ability of these cells to expand with high efficiency in culture must be systematically evaluated.

Several different studies have attempted to induce the differentiation of MSCs from different sources towards dopaminergic neurons to assess their ability to be used in cell based therapies for neurodegenerative diseases such as Parkinson's disease (*Adib et al., 2015*; *Tatard et al., 2007*; *Trzaska, Kuzhikandathil & Rameshwar, 2007*). A wide array of small molecules, cytokines and neurotrophic factors, have been used in different differentiation protocols. For instance, brain and glial derived neurotrophic factors, FGF-8, SHH, cAMP, DAPT have been utilized frequently in neuronal induction (*Adib et al., 2015*; *Tatard et al., 2007*; *Trzaska, Kuzhikandathil & Rameshwar, 2007*).

Here, we assessed the ability of the generated NSC-like cells to differentiate into dopaminergic neurons. Quantitative Real-time PCR and immunostaining of some dopaminergic-specific markers revealed the higher differentiation potential of WJ derived cells towards the neural lineage compared to AT and BM derived MSCs. It is noteworthy that the utilization of different differentiation protocols, sample pool, as well as different analysis tools might lead to significant variation in the differentiation results. Accordingly, a large-scale study with in depth analysis is required in order to verify the exact potential of each MSCs type.

## CONCLUSION

In this study, we report that MSCs derived from adipose tissue, bone marrow and Wharton's jelly can be induced to differentiate into neuron-like cells and further matured into dopaminergic-like phenotype. WJ-MSCs showed a higher neuronal differentiation potential compared to AT and BM derived MSCs. The differentiation of MSCs into neural cells might be a realistic goal as evident by the expression of some neuronal markers after cellular induction. However, it is still early to claim that these generated neuronal cells can be used in cell-based therapies, especially that such differentiation protocols dictate that these cells must cross mesodermal lineage towards a neuroectodermal lineage. The efficiency of all trans-differentiation protocols is still debatable and must be comprehensively analyzed *in vivo* to confirm the terminal differentiation potential.

### Funding
This work was supported by the deanship of scientific research University of Jordan Grant No. 2000. The funders had no role in study design, data collection and analysis, decision to publish, or preparation of the manuscript.

### Grant Disclosures
The following grant information was disclosed by the authors:
Scientific research University of Jordan: 2000.

### Competing Interests
The authors declare there are no competing interests.

### Author Contributions
- Nidaa A. Ababneh and Ban Al-Kurdi conceived and designed the experiments, performed the experiments, analyzed the data, prepared figures and/or tables, authored or reviewed drafts of the paper, and approved the final draft.
- Fatima Jamali and Abdalla Awidi conceived and designed the experiments, authored or reviewed drafts of the paper, and approved the final draft.
## Ethics

The following information was supplied relating to ethical approvals (i.e., approving body and any reference numbers):

an International Review Board (IRB/7/2019) approval at the University of Jordan/Cell Therapy Center (CTC).

## Data Availability

The gene expression analysis data is available in the Supplemental Files.

## Supplemental Information

Supplemental information for this article can be found online at http://dx.doi.org/10.7717/peerj.13003#supplemental-information.

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
