# Peer review of "A comparative study of the capability of MSCs isolated from different human tissue sources to differentiate into neuronal stem cells and dopaminergic-like cells"

_PeerJ, doi:10.7717/peerj.13003_

## Round 0.1 · original submission · Major Revisions

As you will see, the view of this reviewer, which I share, is largely positive, with some caveats which I would like you to address in revision.

We need clarification on the following:
- patient donors - which samples are being used when and how many
- n number and stats on all graphs

Also, I would advise doing a little bit extra on the key outcome - the ability of these MSC sources to differentiation into neuronal-like cells. Whilst you may not be able to do functional assays (which would be the ideal outcome), the conclusion is based on immunofluorescence of markers at 30 days, where the analysis graph has no stats. Hence, further support of this is needed, by, e.g. in cell western or better still, western blots at a few time points. Also, taking the cells out to further differentiate them would generate some interesting extra data if this is possible.

Please address all points in your revision, and I hope you find these comments helpful.

·

Basic reporting

The manuscript is well written in the main. The introduction is logical and leads the reader though to the main aims. There are a few points which I think need clarification. For example, giving the use of iPSCs negatively, when they are actually an excellent opportunity to replace or compliment the use of MSCs. Perhaps to strengthen their example, they could mention the use of retroviruses to generate iPSCs.

The description of differentiation techniques used to direct down a neuronal lineage is well written, but may be better in table form, as it reads rather like a list at present. A table will let readers compare the differs methods use and the the different cell types used.

Experimental design

The methods are clear and concise. Can the authors please clarify the number of donors and which tissue samples were used from donors, as presently it is not clear (e.g. lines 107 and 109). Are the authors aware of the donar age, gender and medical history - these often bias results (but it is understandable if this is unknown)?

The differentiation media used was commercial rather than in-house (it is a shame not to see the ingredients used to initiate differentiation). The differentiation analysis was routine, with correct markers etc, whilst cell proliferation used the MTT assay (which actually measures cell metabolism).

Validity of the findings

The results are mainly well presented with clearly annotated figures and figure legends.

It may have been interesting to collage some images from figure 1 (differentiation staining) to generate a graph so indicate any differences, rather than just rely on a single representative image.

For figure 2, can the authors (i) have the 24 hours on the same scale as the subsequent two time points (currently it is a smaller scale), and (ii) reduce the scale size of the MTT graphs (so they do not appear stretched vertically)?

Figure 4 is nicely presented - but maybe keep the colour scheme for BM, AT & WJ the same throughout the restyle section (rather than changing the colours with each figure)? This makes it a more consistent read. Also, were stats on the image analysis needed in the graphs (n=3 experiments, but they need to qualify how many microscope images were taken to get the graph data).

Figure 5 & 6 - again, nicely presented, but similar to figure 4 points above.

Additional comments

The results are relatively interesting, but quite limited in scope. Neuronal differentiation via dual SMAD inhibitors is fine, but are there other methods which could be compared and the best method optimised?

There is plenty of opportunity to further test the neuronal differentiation, aside from morphology (light microscopy - any chance to quantify the cell extensions etc graphically?) and neuronal cell markers, to define better the various MSC's ability to generate functional cells. Line 282-285 summarises this - that these cells are intermediate, but further differentiation should be studied - why not in this paper, taking it our longer term?

Also, it would be nice to see an explanation of why there is much higher levels of nestin, PAX6, LMX and Sox2 gene expression with WJ, compared, for example, to BM. However when looking at the corresponding immunofluorescence, these is not observed (the levels are the same).

The conclusion that WJ differentiates better into neuronal-like cells is a bit limited, whilst neuronal markers (immunofluorescence) do seem a bit higher, there are no stats of significance attributed to the immune graphs to clarify any difference observed. Also, as this is a key outcome, It is a shame not to see more protein data, such as a Western blot to bearer identify this.

---

## Round 0.2 · Minor Revisions

Please update the manuscript with details of ages and gender of the donors. once this is complete I will be happy to accept.

·

Basic reporting

Resubmission - the paper it well written on the whole. Minor text corrections have been made.

Experimental design

Resubmission - all fine. However please do date state the donor age and gender (in the resubmission comments authors state that they can find this in the records easily).

Validity of the findings

Resubmission - again, this is fine. Suggested extra work has not been carried out (mainly due to funding and covid) - this is unfortunate as it would significantly enhance the paper.

Additional comments

Resubmission - the authors have addressed some items, but missed some extra work. Please add the donor age and gender (as above).

---

## Round 0.3 · accepted · Accept

Thanks for addressing the minor changes. I am now happy to recommend acceptance.